# Effect of Temperature and Water Absorption on Low-Velocity Impact Damage of Composites with Multi-Layer Structured Flax Fiber

**DOI:** 10.3390/ma12030453

**Published:** 2019-02-01

**Authors:** Yiou Shen, Junjie Zhong, Shenming Cai, Hao Ma, Zehua Qu, Yichun Guo, Yan Li

**Affiliations:** 1School of Aerospace Engineering and Applied Mechanics, Tongji University, 1239 Siping Road, Shanghai 200092, China; syoeva@tongji.edu.cn (Y.S.); tjeco_c@163.com (J.Z.); caism0911@163.com (S.C.); haoma@tongji.edu.cn (H.M.); guoyichun19961010@163.com (Y.G.); 2Department of Macromolecular Science, Fudan University, 2005 Songhu Road, Shanghai 200433, China; quzehua@fudan.edu.cn; 3Key Laboratory of Advanced Civil Engineering Materials, Ministry of Education, Tongji University, Shanghai 200092, China

**Keywords:** plant fiber, multi-layer, low-velocity impact, temperature, water absorption

## Abstract

Temperature and moisture can cause degradation to the impact properties of plant fiber-based composites owing to their complex chemical composition and multi-layer microstructure. This study focused on experimental characterization of the effect of important influencing factors, including manufacturing process temperature, exposure temperature, and water absorption, on the impact damage threshold and damage mechanisms of flax fiber reinforced composites. Firstly, serious reduction on the impact damage threshold and damage resistance was observed, this indicated excessive temperature can cause chemical decomposition and structural damage to flax fiber. It was also shown that a moderate high temperature resulted in lower impact damage threshold. Moreover, a small amount of water absorption could slightly improve the damage threshold load and the damage resistance. However, more water uptake caused severe degradation on the composite interface and structural damage of flax fiber, which reduced the impact performance of flax fiber reinforced composites.

## 1. Introduction

In recent years, plant fibers have been utilized as reinforcements to replace asbestos and fiberglass in composites for various applications due to their distinctive characteristics, such as sustainability, biodegradability, abundance, cost savings, lower density, high specific strength, modulus, acoustic absorption, and good energy absorption properties [1,2,3]. However, plant fiber-based composites are mainly limited to non-structural or semi-structural applications due to lack of confidence in their mechanical performance.

The occurrence of impact damage during the service life of fiber reinforced plastic composites is unavoidable and is becoming one of the most important aspects that inhibits their widespread use in industry [4,5]. Among a number of basic mechanical failure modes happening in fiber reinforced composites during impact events, matrix cracking and delamination are the most common damage mechanisms, and are defined as the initial damages of synthetic fiber-based composites. These damages are barely visible to the naked eye but can result in a serious decline in structural stiffness, stability, and load-carrying capability. Therefore, it is important to determine the onset of impact damage within composites for composite design criteria and applications. Damage threshold load (DTL) is observed in synthetic fiber-based composites by examining load–time histories from numerous low velocity impact tests, to identify the impact load at which detectable damage is initiated, as reviewed by Abrate [6,7,8]. Load drops on the impact force vs time plot are often taken to be indicative of strain energy release due to damage propagation within the plate, such as delamination and fiber breakage [9,10,11]. It was found that the DTL of synthetic fiber reinforced composite materials can be affected by many factors, such as projectile diameter, plate dimensions, test temperature [12], as well as impact location [13]. Although plenty of research has focused on low velocity impact performance of plant fiber reinforced composites [14,15,16], the impact damage initiation mechanisms of these composites at various temperatures or water absorptions are still unclear. In order to implement plant fiber composites in high performance structural applications, where low velocity impact can be expected, the onset of the damage, impact response, damage mechanisms, and the key influencing factors of the plant fiber-reinforced composite must be studied.

Previous work [17] has found that plant fibers possess multi-layer microstructures, and each elementary fiber is made up of a primary cell-wall and a secondary cell-wall. The secondary cell-wall is composed of three layers, namely S1, S2, and S3 layers. Most of the cellulose microfibrils locate in the S2 layer, and cellulose is a type of polymer material. The S2 layer consists of numerous cellulose microfibril sublayers (L1, L2… Ln), which run parallel to each other and form a microfibril angle (MFA) with the fiber direction, and the impact properties of the plant fiber composites are closely related to the MFA of the S2 layer [18]. At the center of the elementary fiber, there is a small open channel called a lumen. This multi-layer structure characteristics of plant fibers could cause different mechanical problems, and damage mechanisms of their composites when being subjected to foreign impact loads. Moreover, the chemical compositions of plant fibers are complex, and are composed of hydrophilic cellulose and other organic materials, including hemicellulose, lignin, and pectin. This could cause sensitivity to high temperature and low durability to moisture of the composite. Heat is applied during some of the manufacturing processes of fiber reinforced composites, including hot press, autoclave, and pultrusion. This might have serious effects on plant fibers if the processing temperature is not well controlled [19]. Ma et al. [20] found that a thermal exposure on flax fiber at 180 °C caused thermal decomposition and changes in chemical composition of fiber, which led to a decline in tensile strength of both fiber and its reinforced composite. On the other hand, the mechanical properties of the polymer matrix are highly temperature dependent; it moves from a hard, glassy state to a soft, rubbery state as temperature increases, and the modulus can be several orders lower [21]. As mentioned above, plant fiber is composed of organic polymer materials; the mechanical behavior of its reinforced polymer composites at various temperatures also needs to be studied carefully, such that using them at unfavorable temperatures could be avoided. Opposed to man-made fibers, plant fibers possess a large amount of hydroxyl groups and contain lumen structures, which make it very easy for them to absorb moisture; furthermore, the saturated water absorption is much higher than those of man-made fibers [22,23]. Li and Xue [24] found that exposing plant fiber reinforced composites in a wet environment, particularly in water, for a period of time resulted in a loss of mechanical properties due to the degradation of plant fibers and weakening of the interface.

The impact on composites is a complex problem; the complexity of the microstructure and chemical composition of plant fibers bring new damage mechanisms and scientific problems. This study aims to investigate the effect of manufacturing process temperature, environment temperature, and water absorption on the low-velocity impact response and damage mechanisms of flax fiber reinforced epoxy plastic (FFRP) laminate at the damage initiation level. The failure mechanisms under various circumstances are also revealed with the aid of an optical microscope and scanning electronic microscope.

## 2. Materials and Methods

### 2.1. Materials

The unidirectional flax fabric was supplied by Lineo Co. Ltd, Belgium with a fiber density of 1.50 g/cm^3^, an areal density of 200 g/m^2^, and a thickness of 0.18 mm. Bisphenol-A-based epoxy resin, anhydride-based curing agent, and tertiary amine-based accelerating agent were supplied by Shanghai Zhongsi Industry Co. Ltd, China.

### 2.2. Laminate Fabrication

A hand lay-up method was implemented to fabricate flax fiber reinforced epoxy laminates. Twenty-four layers of flax fabric were stacked up in sequence of (0/90)_6s_ after impregnation with a mixture of epoxy resin and curing agent and accelerating agent (weight ratio of 100:80:1). The impregnated fabrics were then placed in a picture frame steel mold and preheated at 90 ℃ for 1 h and post cured at 120 ℃ with a constant pressure of 1.2 MPa for 2 h before cooling slowly to room temperature. In this study, five temperatures, including 120, 140, 160, 180, and 200 °C were applied on post cured composite laminates. The corresponding composites were denoted as PC120, PC140, PC160, PC180, and PC200, respectively. The resultant composite laminates each had a nominal thickness of 5 mm and an approximate fiber volume percentage of 65%. The test samples were cut from the manufactured composite plates using a slitting wheel. Five samples with dimensions of 25 $\times \;25\;\times \;5\;{\mathrm{mm}}^{3}$ were inspected with the Micro-CT imaging system (SkyScan 1176, Bruker Corp., Billerica, MA, USA), and the porosity of the composites were all below 1%, as shown in the 3D and cross section scanning images of Figure 1. Square samples with dimensions of 100 × 100 mm^2^ were then cut for impact tests.

### 2.3. Experimental Methods

Fourier Transform infrared spectroscopy (FTIR) was performed at room temperature using a Nicolet 6700 spectrometer (Thermofisher, USA) on finely powdered samples (NOPC (no post cure), PC120, PC140, PC160, PC180, and PC200). The powdered samples, mixed with KBr in a weight ratio of 100:3, were molded into pellets using a press machine.

The storage modulus and Tanδ of FFRP laminates (post cured at 120 °C) were studied by using a dynamic mechanical analysis machine (Q800DMA, TA Instruments). The testing temperature ranged from 25 to 250 °C.

Drop weight impact (DWI) tests were conducted using an INSTRON CEAST 9350 drop-weight tower with a 2.27 kg steel carriage, and a detachable hemispherical steel indenter having nominal diameters of 16 mm was employed during testing. All the FFRP samples were placed on a circular ring support (76 mm in diameter) and clamped with a same size ring with 200 N force.

The effect of exposure temperature on the impact damage threshold of the composite was investigated by preheating laminates in the environmental chamber of the impact machine for 2 h prior to the impact test. Here, a series of temperatures ranging from room temperature (23 °C) to 140 °C were applied on preheated samples. The corresponding samples were denoted as TT23, TT40, TT60, TT80, TT100, TT120, and TT140, respectively.

Considering the hydrophilic property of the plant fiber, the effect of water absorption on the impact response of flax fiber reinforced composite laminates manufactured with a post curing temperature of 120 °C were studied here. The water absorption measurement was conducted according to ASTM-D5229. Five rectangular composite samples with dimension of 25 × 25 × 3 mm^3^ were dried in the climate chamber at a temperature of 105 °C until no relative humidity could be measured. Then the samples were soaked in deionized water at 23 °C. The amount of water absorption (W) of the composites was calculated according to Equation (1).
(1)$$W=\frac{W_{t}-W_{0}}{W_{0}}\times 100\%$$
where *W*_0_ and *W*_t_ denote the original weight and the weight after soaking for t time (t = 1 to 16 weeks) of the specimen, respectively. Then, samples with dimensions of 100 × 100 × 5 mm^3^ were soaked in the deionized water at room temperature for seven immersing times, these being 1, 2, 4, 6, 8, 12, and 16 weeks. The corresponding samples were denoted as WA1W, WA2W, WA4W, WA6W, WA8W, WA12W, and WA16W, respectively. Three samples were prepared for each immersing time. After immersing, samples were taken out and dried for the impact test.

The failure area and cross section of the test specimens were inspected with a scanning electron microscope (PHILIPS XL30 FEG, PHILPS, Eindhoven, The Netherlands) and optical microscope (10XB-PC, Shanghai Optical Instrument Factory, Shanghai, China). Table 1 summarizes the information of FFRP samples used in the DWI tests.

### 2.4. Damage Threshold Determination of the FFRP Laminate

The damage threshold load of flax fiber reinforced composites was determined with an experimental method. Firstly, a series of incident energies, including 1, 3, 5, and 6 joules, were applied on the FFRP laminates at room temperature to reveal the impact responses and damage modes. Figure 2 presents contact force–time and contact force–displacement histories of FFRP laminates. It can be seen from Figure 2a that increases in incident energy led to the enhancement of peak contact load and a decline in contact duration. A peak impact load of 3450 N was recorded for specimens impacted with 6 joules of energy for 0.97 ms, after which a sudden drop was recorded. Figure 2b shows the load–displacement curves of FFRP following different impact energies. It can be seen that the loading and unloading curves did not coincide even for the very low impact energy of 1 joule, suggesting small portion of energy was absorbed through internal damage of the specimen. As for the incident energy of 1 joule, linear behavior was found in the beginning of the load–displacement curve, then the slope slightly changed near the peak load, as shown with the red dashed lines; this indicates that the small amplitude reduction in laminate stiffness was caused by intraply transverse matrix cracking. The load–displacement curve of the specimen impacted with 6 joules is nearly identical to that of 5 joules for loads up to the damage threshold load. More energy was absorbed, and larger indent displacement and damage area were observed for the FFRP laminate impacted with 6 joules. It was proved by Sjoblom et al. [25] that the matrix cracks do not significantly affect the overall laminate stiffness during an impact event, but the matrix crack tips are the initiation points for delamination and fiber breaks. However, delamination and fiber breakage can dramatically change the local or global stiffness of the composite and affect the load–time response. Therefore, the force–time and force–displacement history suggested no delamination and fiber breakage occurred at and below the impact energy of 5 joules.

The samples were visually examined after being impacted with energies of 5 joules. Circular indentation pits with diameters of 3 mm and dent depths of less than 0.5 mm were observed on the top surface, while matrix cracking caused by a high tensile bending force was observed on the bottom surface, as shown in Figure 3. The matrix cracks were along the orientation of fibers in the unidirectional layers due to a property mismatch between the fiber and matrix, which led to stress concentrations at the fiber–matrix interface. The samples subjected to 5 and 6 joules of impact energy were sectioned slightly off the impact center and mounted in epoxy resin; cross sections of the samples were carefully polished to the impact zones, and with the aid of an optical microscope, the damage mechanisms were revealed. As seen in Figure 4a, matrix cracks were found in flax fiber yarns and initiated from the bottom of the laminate with 5 joules of impact energy. Delamination appearing in the adjacent interlayers nearby the bottom surface was caused by the mismatch of fiber directions within two adjacent plies. In terms of 6 joules of impact energy, fiber breakage was observed in Figure 4b. This indicates that 5 joules were the crucial incident energy causing initiation damage for this type of composite. The force to initiate damage was taken as the maximum value in the load–time trace during a test in which matrix cracking and delamination had just initiated.

## 3. Results

### 3.1. The Effect of Manufacturing Process Temperature on Damage Threshold

The variations of chemical components of flax fiber treated at temperatures of 120, 140, 160, 180, and 200 ºC were analyzed using FTIR spectroscopy, as shown in Figure 5. The infrared absorption spectra demonstrate that the peak intensities at 1608, 1247, 1184, 1034, 865, and 831 cm^–1^ gradually decreased with the increase of treatment temperatures; a serious decline occurred at the treatment temperature of 200 ºC. The six peaks mentioned above corresponded to different functional groups of certain chemical components within flax fibers, as shown in Table 2. The varieties of the intensities of these absorption peaks indicated that high treatment temperatures could seriously change or destroy the chemical constitution of plant fibers, such as hemicelluloses, pectin, lignin, and partial unstable cellulose. The DTL of FFRP manufactured at various post curing temperatures was obtained and is shown in Figure 6. It can be seen from the bar chart that the DTLs of PC120, PC140, and PC160 were similar; significant declines on DTL were found for PC180 and PC200: the reductions were 31% and 37%, respectively. It can be concluded from the results that enhancing the manufacturing process temperature leads to a reduction of the damage threshold and earlier damage on the plant fiber-based composites. This is due to changes in the chemical composition of plant fibers when temperatures are above 160 °C, evidenced with the aid of FTIR spectroscopy, as shown in Figure 5.

### 3.2. Effect of Exposure Temperature on Damage Threshold

It is well known that the resistance to plastic deformation, strain to failure, and fracture toughness of polymer matrices are strongly dependent on temperature as a consequence of their viscoelastic natures. Since initial damage in the FFRP laminate involved fracture on the matrix and delamination, it is likely that the exposure temperature influences the failure threshold of the FFRP laminates. Figure 7 shows that the storage modulus of the FFRP starts to drop from 100 °C, and the glass transition temperature (T_g_) of FFRP is 125 °C. Therefore, a range of test temperatures from 23 to 140 °C was chosen for the impact tests in this study. As stated before, this temperature range has no significant influence on the chemical composition of flax fiber. The effect of temperature on the matrix of FFRP was the focus of this part of the research.

A clear trend of gradually decreasing DTL was established with increasing temperature over the range of 23 to 140 °C for FFRP laminates following an impact energy of 5 joules, as seen in Figure 8. It can be seen that the threshold load dropped 15% when the temperature was above 100 °C, and the DTL value dropped 50% at 140 °C compared to that at room temperature, which was consistent with the decline trend of the storage modulus. Close observation of the rear side of these samples showed fiber/matrix debonding and flax fiber structural damage, including elementary fiber splitting, cell-wall peeling, and microfibril bridging, as shown in Figure 9.

### 3.3. Effect of Water Absorption on Damage Threshold

Figure 10 shows the water absorption and the DTL of FFRP as a function of the square root of the immersion time at room temperature. From the moisture–time history, it can be seen that the water uptake displayed linearity to the square root of immersion time for the initial four weeks, then became steady from 4 to 12 weeks; no significant increase in the water absorption of WA16W samples was found compared to that of the WA12W samples, meaning that the water absorption of FFRP tended to be stable after 12 weeks immersion at room temperature. In the first stage of water absorption for FFRP, water molecules permeated into composites through different absorption and combination patterns due to the complex components and multi-layer structure of the plant fiber. The microfibrillar angle was modified when free hydrophilic groups in the S2 layer were attached onto the water molecules [23]. This enhanced the tensile strength of fibers to some extent, thus increasing the DTL of the composites at the initial stage of the water absorption. As more water was absorbed by the specimens, larger swelling stress caused matrix cracking around the swollen fibers, and more tunnels were generated for water absorption, as shown in Figure 11, which led to the degradation of the fiber–matrix interface and poor stress transfer efficiencies. It was found by Li and Xue [24] that the reduction on interlaminar shear strength (ILSS) of FFRP was approximately 30% for an immersion time of 17 weeks. In this study, the DTL decreased approximately 9% with an immersion time of 16 weeks due to the combined effects of an increase in fiber yarn tensile strength and a decline on the ILSS. Figure 12 shows serious fiber splitting, accompanied by cell-wall peeling and elementary fiber fracture on FFRP under impact damage initiation. This indicates that the water absorption not only caused interface degradation between fiber and matrix, but also destroyed the structure of the flax fiber; delamination was induced between different layers inside of the flax fibers (S1, S2, S3 layers), and this resulted in the cell-wall peel off from the fiber.

## 4. Discussion

Assessing the effect of temperature on the DWI test of composite materials is extremely important due to the fact that manufacturing temperatures or expose temperatures during maintenance operations are moderately higher than the ambient temperature. It is evidenced from Figure 5 that the chemical component for flax fiber seriously changed above 160 °C. Cellulose, hemicellulose, lignin, and pectin are the main polymers in plant fibers like flax. At temperatures above 160 °C, the degradation of these polymers (mainly hemicelluloses and amorphous parts of cellulose) may occur and the chemical structure of lignin could change, which consequently decreases the load capacity of flax fiber to a large extent [26]. Previous work [20] has proved that there was no significant influence on curing degree and tensile properties of the epoxy by enhancing post curing temperature. Therefore, load capacity reduction on flax fiber caused by elevated temperature was the main reason for the declined DTL of FFRP due to fiber tension on the rear face that dominated the impact damage force. On the other hand, since water molecules in plant fiber played an effective role on plasticizing the fiber [27], the removal of water molecules within the flax fiber above 150 °C also led to reduction of fiber toughness. Therefore, the PC180 and PC200 laminates fractured in a brittle manner. Owing to the fact that the compositions of most plant fibers are similar, the manufacturing process temperature should be controlled at or below 160 °C, as exorbitant temperature will damage the plant fiber microstructure and lead to significant reduction on the DTL and the impact resistance of their reinforced composites.

Increasing the exposed temperature of FFRP also causes the reduction of the DTL due to the decreased stiffness of the FFRP composite and the increased ductility of the laminate, which causes the degradation of the interface between layers, and debonding between fiber and matrix. This results in ineffective stress transfer and deterioration of impact resistance. For the amorphous polymers in plant fibers, enhancing the temperature leads to the increase of molecular thermal motion energy, and chains within the polymers would rotate to some extent; thus, these polymers would also become soft and ductile. Therefore, the cohesion between different cell-wall layers, as well as microfibrils, become weak, resulting in fiber structure damage and reduction of the fiber bearing capacity. It can be concluded that the plant fiber-based composites can maintain favorable performance at exposure temperatures below 100 °C.

It is interesting to notice that the DTL of FFRP increased with immersion in water for one week. This enhancement is due to the reason that when free hydroxyl groups in the amorphous parts of the cell-wall polymers of plant fibers attach onto water molecules, the MFA is modified and leads to an increase of the tensile strength of fibers after water taking, which was proved by Astley and Donald [28] using an X ray diffraction (XRD) technique. The MFA is increased due to the distinct absorption and binding degree between components in different cell-wall layers of flax fiber with hydrone. Since during an impact process membrane stiffening played a significant role and resulted in a tension at the bottom surface of the plate, the improvement of the tensile strength of flax fibers resulted in the enhancement on the DTL of FFRP. Moreover, previous work [24] has found the interlaminar shear strength of the FFRP also increased in the first immersion week followed by a continuous decline with more water absorption. This is due to an expansion of flax fibers that is much larger than that of epoxy; this exerts a radial stress on the matrix. At the initial water absorption stage, only a small amount of water was absorbed by flax fibers within FFRP; this resulted in a smaller hygroscopic stress and enhanced the interfacial clamping and frictional stresses, which led to an increased ILSS of FFRP. These reasons caused the improvement on the DTL of FFRP laminates at the first immersion week. However, more water uptake and increased immersion time led to degradation of plant fibers and the interface of the composite, which left voids and cracks within the composite, as shown in Figure 11. Li and Xue [24] monitored the soaking solution of FFRP using the FTIR spectra and found that pectin, hemicelluloses, and some poorly crystallized celluloses inside of flax fibers leached out. Moreover, different swelling behaviors of each cell-wall layer induced the fiber splitting and cell-wall peeling as shown in Figure 12. The above damage mechanisms result in the reduction on the DTL of FFRP. Therefore, it can be concluded that the effect of the water absorption is not significant on the impact performance of plant fiber reinforced composites in the beginning of the water uptake process; a small amount of water absorption (in this study, the value was 4.2%) can improve the damage threshold load. Deterioration on impact performance was found at higher water absorption (in this study, the value was over 6.2%) due to the degradation of plant fiber structure and the interface of composites.

## 5. Conclusions

In this study, the low-velocity impact response and damage initiation mechanisms of flax fiber reinforced epoxy laminates under certain temperature and water absorption conditions were evaluated experimentally. The low-velocity impact response and damage initiation mechanisms of plant fiber reinforced composites, especially under elevated temperature and moisture relevant circumstances, reveal the difference with synthetic fiber-based composites, which manifests as fiber chemical composition changing, and degradation of the fiber/matrix interface of composites and cell-wall layer interfaces of plant fibers. Damage mechanisms, including fiber splitting, cell-wall peeling, and microfibril bridging, were observed owing to the multi-layer microstructures of plant fibers. Moreover, suggestions on manufacturing parameters and potential service conditions of plant fiber reinforced composite structures are provided, which might be helpful to the structural integrity and design of this type of composite by avoiding unfavorable conditions. The manufacture post curing temperature for plant fiber-based composites should be controlled at or below 160 °C in order to avoid reductions of the impact resistance due to the decomposing of fiber chemical composition and microstructure destruction at elevated temperatures. The service temperature for FFRP should be lower than 100 °C, since the DTL of the laminate will dramatically decline due to the ductility increase of polymer materials under higher temperatures, which cause degradation of the interface of both the fiber/matrix and different cell-wall layers. Similar to the effects of temperature, too much water uptake also causes severe degradation of both interfaces of composite and cell-wall layers of plant fibers.

## Figures and Tables

**Figure 1 materials-12-00453-f001:**
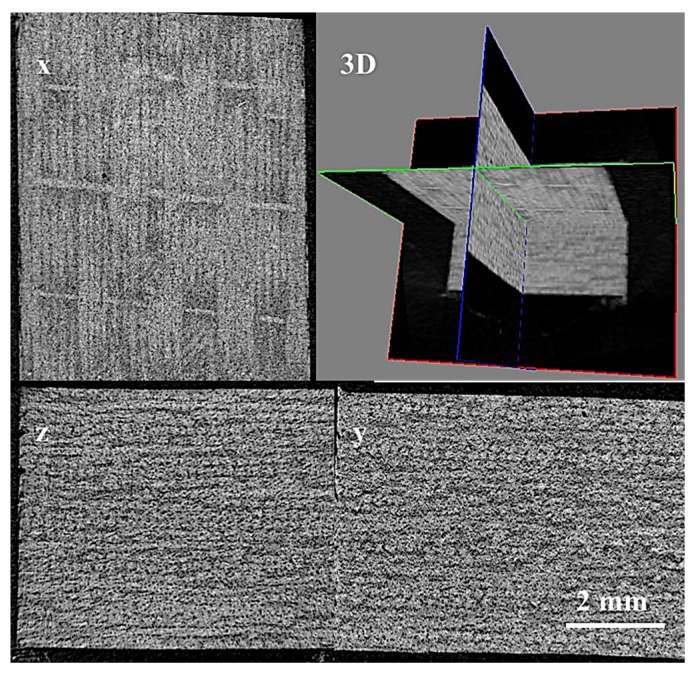
The 3D and cross section scanning images of the PC120 sample.

**Figure 2 materials-12-00453-f002:**
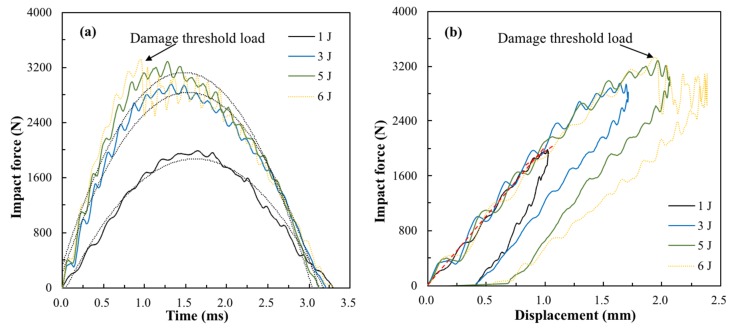
(**a**) Impact load–time curve and (**b**) impact load–displacement curve of flax fiber reinforced epoxy plastic (FFRP) following incident energies of 1, 3, 5, and 6 joules.

**Figure 3 materials-12-00453-f003:**
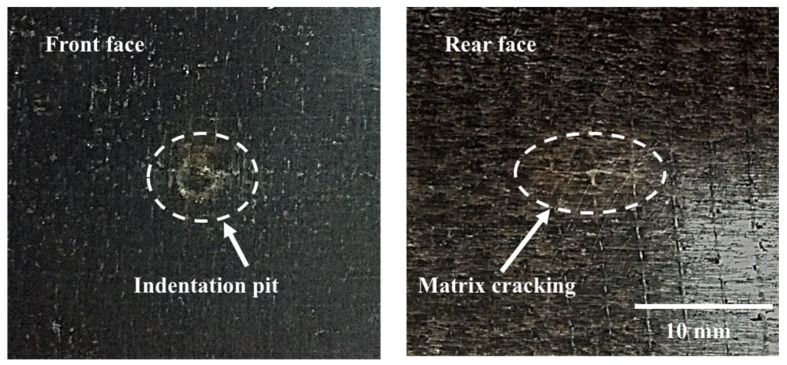
The appearances of FFRP laminate following an impact energy of 5 joules.

**Figure 4 materials-12-00453-f004:**
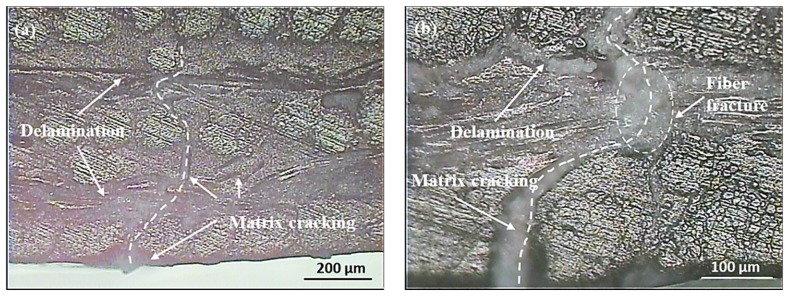
Micrographics of cross-sections for FFRP laminate subjected to (**a**) 5 joules and (**b**) 6 joules of impact energy.

**Figure 5 materials-12-00453-f005:**
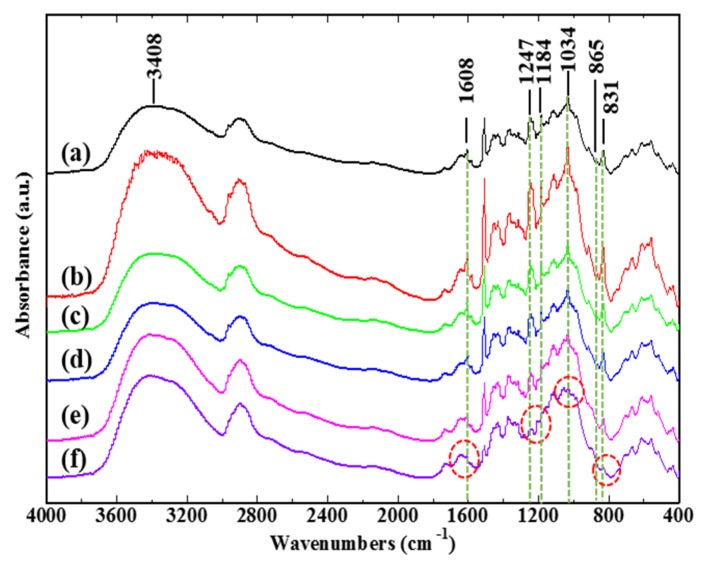
FTIR spectra of flax fiber at (**a**) room temperature, (**b**) 120 °C, (**c**) 140 °C, (**d**) 160 °C, (**e**) 180 °C (**f**) 200 °C.

**Figure 6 materials-12-00453-f006:**
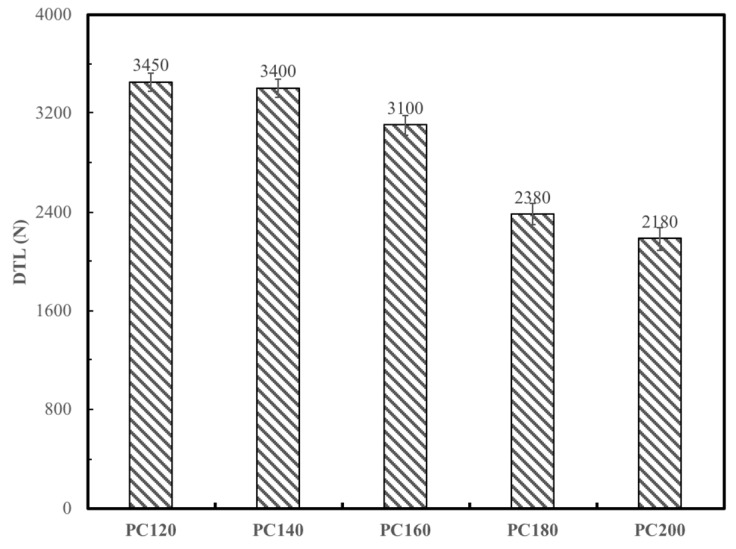
The damage threshold load (DTL) of FFRP laminates manufactured with different post curing temperatures.

**Figure 7 materials-12-00453-f007:**
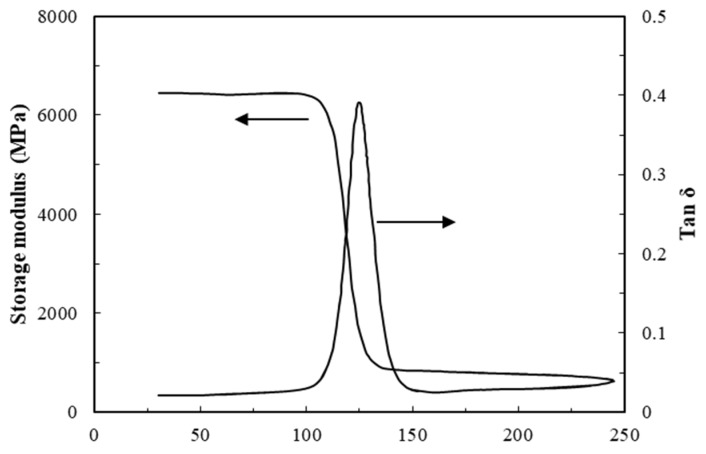
Dynamic mechanical properties of FFRP laminate.

**Figure 8 materials-12-00453-f008:**
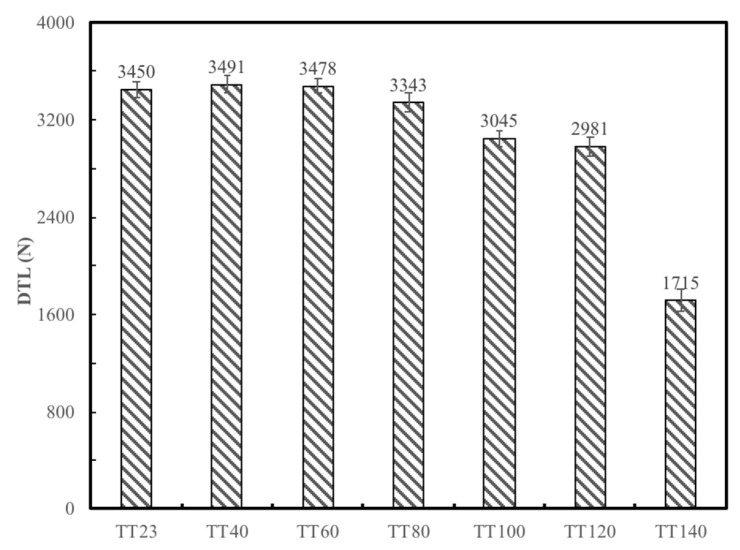
The DTL of FFRP laminates tested at different temperatures.

**Figure 9 materials-12-00453-f009:**
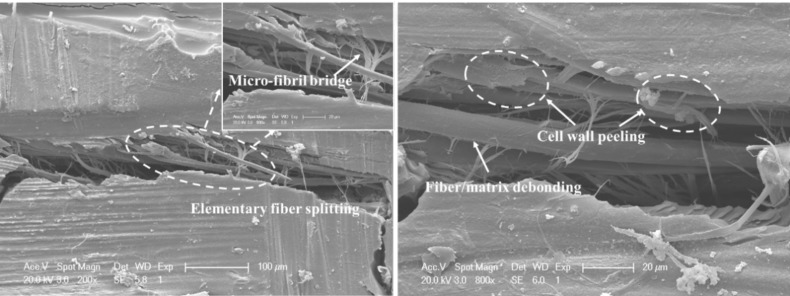
Micrograph of damage initiation modes for the FFRP following impact test at 140 °C.

**Figure 10 materials-12-00453-f010:**
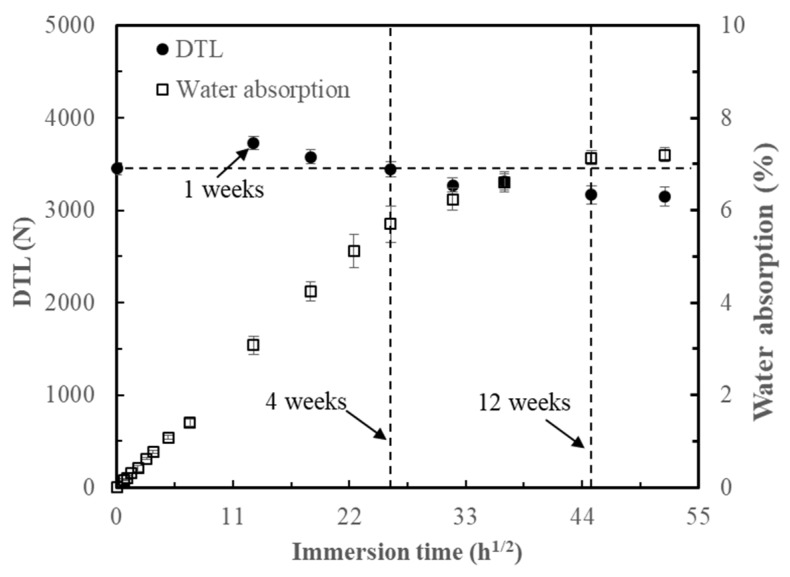
Variations of DTL and water absorption with the square root of immersion time.

**Figure 11 materials-12-00453-f011:**
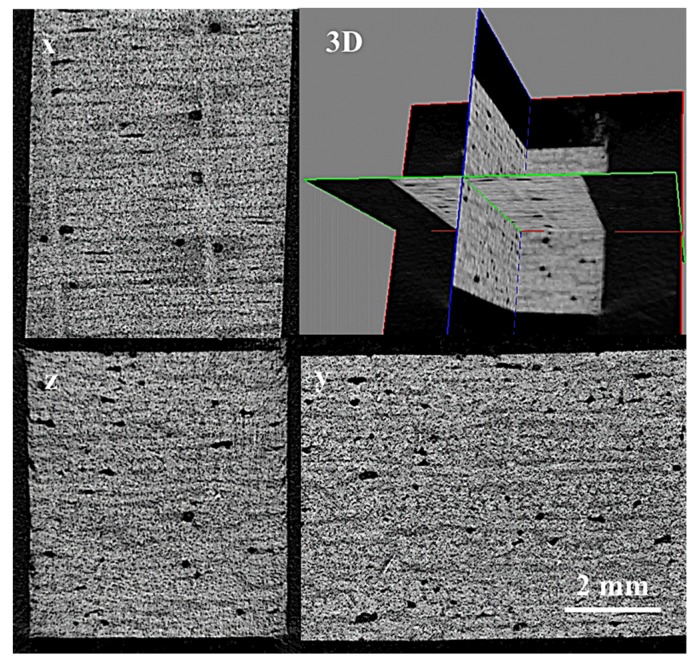
The 3D and cross section scanning images of the WA12W sample.

**Figure 12 materials-12-00453-f012:**
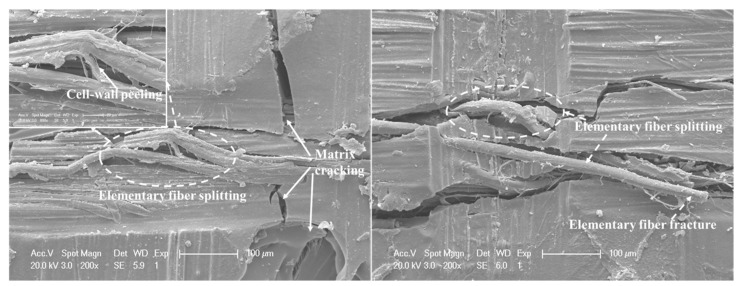
Micrograph of the damage initiation modes for the WA12W sample.

**Table 1 materials-12-00453-t001:** Summary of flax fiber reinforced composite samples for the drop weight impact (DWI) tests.

Name	Post Curing Temperature (°C)	Nominal Thickness (mm)	Fiber Volume Fraction (%)	DWI Test Temperature (°C)	Water Immersion Time (Week)
PC120	120	5.0	65	23	0
PC140	140				
PC160	160				
PC180	180				
PC200	200				
TT23	120	5	65	23	0
TT40				40	
TT60				60	
TT80				80	
TT100				100	
TT120				120	
TT140				140	
WA1W	120	5	65	23	1
WA2W					2
WA4W					4
WA6W					6
WA8W					8
WA10W					10
WA12W					12
WA16W					16

**Table 2 materials-12-00453-t002:** Characteristic bands of the infrared spectra of flax fiber.

Wave Numbers (cm^−1^)	Functional Group	Attribution Component
3408	–OH	Cellulose, hemicelluloses, and lignin
1608	C=C	Lignin
1247	C–O	Lignin
1184	C–O–C	Pectin
1034	C–O	Cellulose and hemicelluloses
865	C–H	Lignin
831	C–H	Lignin

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
