# Peer review of "Effect of Temperature and Water Absorption on Low-Velocity Impact Damage of Composites with Multi-Layer Structured Flax Fiber"

_materials, 2019, doi:10.3390/ma12030453_

Round 1
Reviewer 1 Report
This paper investigates, through low velocity impacts the effects of temperature and water absorption on flax fibre composites.
The results are interesting and scientifically sound. the overall style is good, but I think it could be improved to make it more readable.
General Comments
I would substitute "plant fiber" with "flax fibres" in the title.
Are you writing in British or American English? Be consistent!
You need to seriously correct and review grammar and spelling. I suggest that you get it reviewed by a native English speaker. I started to point out those errors, but it is beyond the scope of a review.
Abstract
Minor rewriting required to improve language and style, e.g. “chemical composition decomposing” sounds quite bad.
Introduction
The literature background is insufficient, you need a more detailed and focused review, functional to the message and conclusions of your work.
You seem to be mixing introduction and results. I am confused from the paragraph between line 65 and 79 and Figure 1, is this you work or is it coming from the literature? If it’s your work the introduction is not the best place for it, if it is from the literature iti is not properly quoted.
You need to reorganise this section to be more consistent and functional to your message.
“plant fiber has been utilized” id use plural in this case.
“due to lacking of confidence” should be “lack of confidence”
“Damage threshold load (DTL) is existed for composites below which no serious damage occurs but above which significant damage is produced, it can be the division of the barely visible impact damage (BVID) and visible impact damage (VID).” I get what you want to say, but the grammar is completely wrong…
“A sudden load drop in the load–time history manifested in the form of matrix cracking, fiber breakage and delamination due to the loss of stiffness from unstable damage development [10,11].” This sentence grammatically makes no sense…
You mention the work from Clemons and Sanadi on kenaf that is irrelevant, you should concentrate only on flax fibres. I’d suggest ot oention the work from Longana et al. ( https://www.mdpi.com/2313-4321/3/4/52 )
Material and Methods
You talk about sourcing and treating separately dry flax fabric and resin in the material section you then talk about “prepreg” in the laminates fabrication. This is not possible, or you worked with prepreg or you worked with dry fibres and resin with infusion/wet lay-up and compaction moulding process…
What is the rational behind the selection of so many different thicknesses? Why did you do it? What were you trying to investigate?
I don’t believe that the fibre volume fraction is 65%, especially with an infusion and compaction process…
Figure 2 is rather poor.
Did you seal the sides of the specimens for the moisture absorption test, can you reference the standard you used?
The sentence between line 150 and line 153 is very convoluted and confusing.
I would suggest showing a “testing matrix” including all the combinations of thickness, exposure temperature, moisture absorption time etc... you used.
You don’t describe any methodology in this section.
Results
Section 3.1 of the Result section is extremely convolute and it’s more descriptive of the methodology rather than the results, I think you can, after reorganising the paper, be more concise and effective in your description.
Sections 3.3 and 3.4 is extremely interesting. Considering those results I would suggest to rewrite the Methodology section to precisely introduce all the characterisation techniques you are using here.
Discussion
This section is adequate, however having a well-defined testing matrix might help in rearranging it to be more effective. I would also try to use graphics and table to summarise the results.
Conclusions
This section is very good.
Author Response
The authors would like to thank the reviewer for his/her constructive comments. Please find the attachment of our responses.

Reviewer 2 Report
This manuscript investigated the effect of post-curing and water absorption on low velocity impact damage of flax fiber-epoxy resin composites. The article itself is interesting and deserves publication, but I still found some items that need to be addressed properly before publishing. The items are mentioned in follow:
Part 2.3 must be changed to water absorption. Please be sure that the term of “moisture content” has been changed to “water absorption” in entire of the text and figure (fig. 12), as the samples were soaked in water.
Why not all results are shown in the figures? You have 5 different composites thickness variations and 5 different post curing regimes (as explained in the material method part) with additional untreated control. Therefore, why you only showed the mechanical strength of thickness variations and not the post cured ones? The same question for water uptake, FTIR, and etc.
Please change the caption of figure 7, as you only performed the post-curing after making composites, and not heat treatment of the flax fiber before use them in the composites.
The authors frequently used FFRP laminates in the results and discussions, which composition do you mean exactly? You have FFRP with different thicknesses and post-curings, therefore, it is not clear about which composition you are talking.
Similar comment, which composition did you choose for DMA analysis? And water absorption?
P12 L324-327, Authors explanation is not precise by saying “The amorphous materials (hemicelluloses, pectin and lignin) existed as adhesives to different cell-wall layers and micro-fibrils in plant fiber, the absence of these binding material leads to damage on the structure of flax fibers, which consequently decreased the load capacity of fiber to a large extent.” Cellulose, hemicellulose, lignin and pectin (generally lower content compare to others) are the main polymers in the natural fibers like flax. At temperature above 160C the degradation of the mentioned polymers (mainly hemicelluloses and amorphous part of cellulose) may occur and the structure of lignin could change. For more information please read following article about wood degradation:
Hosseinpourpia, R.; Adamopoulos, S.; Holstein, N.; Mai, C. (2017) Dynamic vapour sorption and water-related properties of thermally modified Scots pine (Pinus sylvestris L.) wood pre-treated with proton acid. Polymer Degradation and Stability, 138: 161-168.
P13 L349, please change “free hydrophilic groups in S2 layer” to “free hydroxyl groups in the amorphous parts of the cell wall polymers”.
Author Response

(The authors gave the same response as above.)
